# Development of the FE In-House Procedure for Creep Damage Simulation at Grain Boundary Level

**Qiang Xu \*, Jiada Tu and Zhongyu Lu**

School of Computing and Engineering, Huddersfield University, Huddersfield HD1 3DG, UK;
u1361251@hud.ac.uk (J.T.); z.lu@hud.ac.uk (Z.L.)
\* Correspondence: q.xu2@hud.ac.uk; Tel.: +44-1484-472842

**Abstract:** A two-dimensional (2D) finite element framework for creep damage simulation at the grain boundary level was developed and reported. The rationale for the paper was that creep damage, particularly creep rupture, for most high temperature alloys is due to the cavitation at the grain boundary level, hence there is a need for depicting such phenomenon. In this specific development of the creep damage simulation framework, the material is modeled by grain and GB (grain boundary), separately, where smeared-out grain boundary element is used. The mesh for grain and grain boundary is achieved by using Neper software. This paper includes (1) the computational framework, the existing subroutines, and method applied in this procedure; (2) the numerical and programming implementation of the GB; (3) the development and validation of the creep software; and (4) the application to simulate plane stress Copper–Antimony alloy. This paper contributes to the development of finite element simulation for creep damage/rupture at a more realistic grain boundary level and contributes to a new understanding of the intrinsic relationship of stress redistribution and creep fracture.

**Keywords:** creep rupture; creep grain boundary; finite element method; grain boundary cavitation; creep damage; poly-crystal

---

## 1. Introduction

Creep damage is a serious problem limiting the lifetime of high-temperature components in many practical applications. A good understanding and accurate description of creep deformation and creep rupture is of great interest to people who research this field.

It is generally understood that under creep conditions, the higher the temperature, the quicker the deformation and the shorter the lifetime of material. On the other hand, the higher the operating temperature, the higher the thermal efficiency for power plants.

It is generally accepted that, for the majority of metals and alloys, creep rupture is due to creep cavitation at the grain boundary where cavities nucleate, grow, and coalesce [1,2].

Three different approaches are used in the modelling of creep fracture [3]: (1) at the macroscopic level, classical fracture mechanics approaches are extended to time-dependent behavior; (2) still at macroscopic level, in continuum damage models, cavitation is incorporated in an average, smeared-out manner by means of a damage parameter; and (3) the micro-mechanical models where various physical mechanisms are directly involved.

Creep continuum damage mechanics have been developed and widely used now and their applicability depends on the reliability of the development of a set of creep deformation and damage equations and the availability of a computational platform (typically finite element analysis (FEA) package). However, the most current approach in the development of creep damage constitutive equation is phenomenologically based on using macroscopic creep strain to fit models of various

kinds including creep cavitation. In a comprehensive review paper, Xu et al. [4] identified that the current phenomenological approach suffers a lack of precise understanding surrounding the damage, the constitutive model thus has issues of low reliability for extrapolation beyond the stress range which has been calibrated and difficulty in generalizing a one-dimensional constitutive equation to a three-dimensional one.

Cavity nucleation [2]: cavity nucleation mechanisms are still not well understood. It has generally been observed that cavities frequently nucleate on grain boundaries, particularly on those transverse to a tensile stress; agglomeration of vacancies, dislocation pile-up, and grain boundary sliding have all been considered to promote nucleation. It has long been suggested that (transverse) grain boundaries and second phase particles are the common locations for cavities. Empirical equations of nucleation were well established for use.

Cavity growth [2]: research has suggested various cavity growth models [3] including (1) diffusion-grain boundary control; (2) diffusion-surface control, (3) grain boundary sliding, and (4) constrained diffusion cavity growth. The local (true) normal stress has been used as the driving force, hence it is more realistic; and this model has been found in good agreement with experimental observation.

Recently, synchrotron micro-tomography has been used to investigate the cavitation of high Cr steel [5–7], and continuous cavity nucleation and cavity growth models were calibrated by Xu et al. [8] and an explicit creep fracture model, based on the coalescence of grain boundary cavities was derived [8]. The applicability of Xu's creep fracture lifetime model to a stress range of 120 MPa to 180 MPa and a lifetime of 2825 to 51,406 h, has been demonstrated with 87% of accuracy [9].

The micro-mechanical model initially was focused on the single cavity to that of the failure of a polycrystalline aggregate comprising a number of grains [3], however, it is still not feasible to directly incorporate them into any engineering analysis due to the need of computational power. Hence, the need of the development of smeared-out grain boundary constitutive equations for macroscopic creep deformation and damage.

Onck and van der Giessen [3] were amongst these to propose the concept of grain and grain boundary elements, via a two-dimensional version. The material in the grain is assumed to be homogeneous and to deform by power creep law in addition to elasticity. Grain boundary processes like cavitation and sliding are accounted for by grain boundary elements that connect the grains. Results are compared with the full-field finite element analysis, the method is demonstrated to capture the essential features of creep fracture, like constrained cavitation and interlinkage of micro-cracks. It also reported that there is a gain in computational time of 600 times by using the smeared-out element.

This micro-mechanical based smeared-out grain boundary element has eventually been further developed [10,11] for the simulation of copper–antimony alloy, and the main contents are: (1) grain boundary nucleation: Dyson's empirical equation [12] has been consulted; (2) cavity annihilation: probabilistic description of crack annihilation [13] has been adopted; (3) cavity growth: constrained cavity growth model [1,14] adopted; (4) grain boundary sliding: Ashby viscosity model [15] adopted; and (5) creep fracture criterion when the cavity area fraction along grain boundary reached 0.5, experimental observation Cocks and Ashby [16]. In this model, grain boundary sliding has been considered for deformation, but not applied to the cavity nucleation. This proposed model is for 3-dimensional. A slightly simple version has been developed by [17], where the annihilation was not considered, also it is a two-dimensional version.

The grain element has been modelled by simple power law in [3] and [17], both have captured the main features of the creep damage occurring in the grain boundary; sophisticated slip-system model has also been developed and utilized in [10]. Thus, it is justified to adopt simple power law in this type research unless it has been found it is not suitable anymore.

The grain boundary element can be implemented via the cohesive zone element [10,17]; the cohesive zone element can have a small thickness or no thickness at all, such as Goodman element [18], but the

mechanical properties can and have been fully represented through its formulation. The former is slightly more complicated in computation. Hence it is desirable to use Goodman element.

Both the cohesive zone element and contact element can be implemented in ABQUS or in-house software. The published work [13,17] used ABAQUS. The authors of this paper have extensive experience and already developed creep damage software for a multi-material zone version.

It is reported [17] that the mesh size of the grain boundary, assuming a perfect hexagonal shape, that eight grain boundary elements per side is required.

It is concluded that: (1) a computational platform able to model grain and grain boundary separately is of importance for research; (2) a two-dimensional version is not only the first step before the development of three-dimensional version, the two-dimensional version is still of use.

FORTRAN language on the Visual Studio 2013 platform (version 11.0.61219.00, Microsoft, Redmond, WA, USA), which is based on object-oriented programming (OOP) [19], will be used in this software development. This procedure is developed under the framework of the traditional FEM, some subroutines for the assembly and solution the stiffness matrix are from Smith et al. [20] and the non-linear displacement iteration method is from Hayhurst et al. [21]. The specific work is based on some existing subroutines and methods, combined with a series of subroutines which were developed to implement the numerical methods of GB to obtain the FE in-house procedure which can realize the 2D polycrystalline creep simulation. The main purpose of this paper is to record and present the development process, more specifically, it reports the program framework and the theoretical background. Finally, through the bi-grain case study, it benchmarked the numerical stability and accuracy of the entire system. The generation of grain boundary mesh will be achieved by the use of Neper (version 3.3.1, Romain Quey, Cornell University, Ithaca, NY, USA) [22].

In the following sections, this paper will report the specific development of creep damage simulation framework: theory, development, validation, and its application to plane stress test of copper-alloy. This paper contributes to the methodology and new insight of the intrinsic relationship of stress redistribution and failure.

## 2. Overview of the Procedure

The in-house procedure was developed from the FORTRAN language on the Visual Studio 2013 platform (version 11.0.61219.00, Microsoft, Redmond, WA, USA), which is based on object-oriented programming (OOP) [19] method to implement the microscopic simulation of creep. In essence, the procedure solves the creep problem, which is a kind of time-dependent and non-linear boundary problem. For simplicity, only plane strain is considered under the 2D case.

### 2.1. Computational Framework

The flow diagram structure of the in-house procedure is shown in Figure 1, its idea came from program P61 of Smith et al. [20] for homogeneous material and it is expanded into this non-homogeneous version, highlighted in red. These exiting techniques are adopted: the assembly and solution of the global stiffness matrix, the stiffness matrix of the plane strain element, the integration method of the constitutive equation, the nonlinear iterative method, etc. The independent development parts are marked with a red dotted line box and are summarized as follows:

1. Calculation of the GB element stiffness matrix.
2. Generation of the GB creep body loads.

In the following sections, we will introduce the existing technologies and subroutines used in this procedure, and then describe the parts of independent development, from a mathematical perspective and through program implementation.

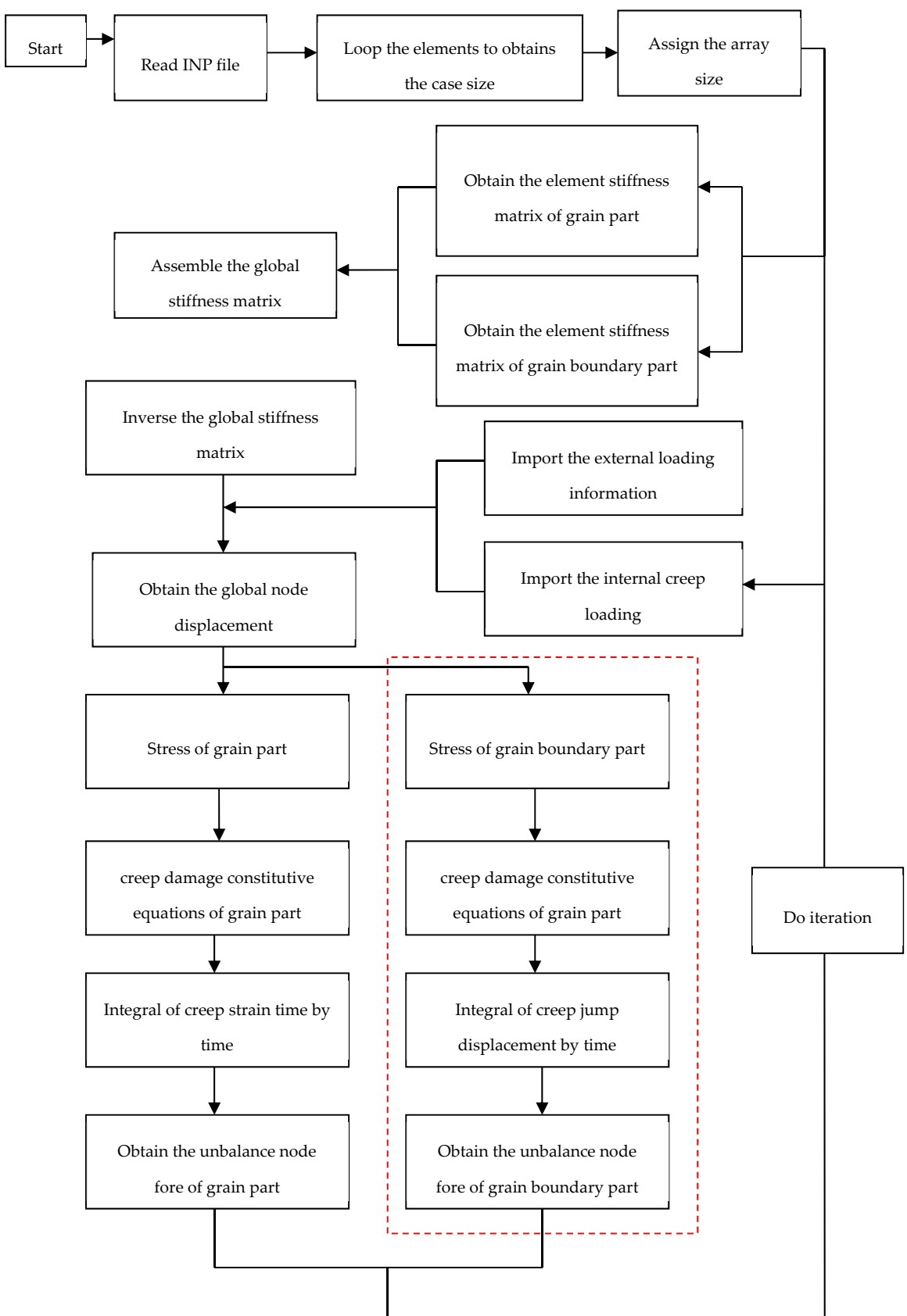

**Figure 1.** Flow diagram structure of the in-house procedure.

### 2.1.1. Packaged Block

The most relevant standard subroutines (11) were adopted from the library 'main' directly [20] and they are listed in Table 1 for completeness, where the description under the function column was simplified by current authors. These subroutines mainly implement the functions of: (1) obtaining the element stiffness matrix; (2) assembling and storing the element to the global stiffness matrix; (3) the solution of the global stiffness matrix, etc. The specific details of these subroutines are summarized in Table 1.

**Table 1.** List of the standard subroutines for the in-house procedure.

| Name | Function |
|---|---|
| formnf | Returns the nodal freedom array. |
| num_to_g | This subroutine is used to obtain the element steering array from the nodal freedom array and element node number. |
| fkdiag | This subroutine is used to store the global stiffness matrix by the lower triangle stored method, it returns the bandwidth value. |
| sample | Returns the local coordinates and the weighting coefficients for the element integration by Gauss method. |
| deemat | Returns the stress–strain matrix. |
| shape_fun | Returns the shape function of each Gauss integrating points. |
| shape_der | Returns the shape function derivatives of each Gauss integrating points. |
| invert | Returns the inverse matrix onto itself. |
| fsparv | Return the global matrix in skyline form, which is from the lower triangular global stiffness matrix. |
| sparin_gauss spabac_gauss | These two subroutines are used to solve the global stiffness matrix, by forward and back-substitution of Gaussian factorized vector of pervious stiffness matrix. |

### 2.1.2. Non-Linear Creep Iteration Method

Creep is a kind of time-related visco-plastic deformation, therefore, the solution of this problem requires non-linear iteration. Here, the displacement iteration method is chosen to solve the residual stress updating. The displacement method was proposed by Hayhurst et al. [21], which was adopted to implement the damage analysis of Weldment [23].

The iterative process for solving this creep problem is as follows. Firstly, calculate the node displacement. This involves multiplying the total load and the inverse stiffness matrix where the total load consists of the actually applied external load plus the additional load at the node due to stress redistribution. Secondly, obtain the new elastic strain for each element. This is achieved by deducting the creep strain from the total strain which, in turn, is obtained by multiplying the displacement with the [B] (Displacement-Strain) matrix. Finally, the newly updated element stress, strain, damage, etc., will be output for next iteration.

### 2.1.3. Generation of the GB Creep Body Loads

The conventional body force from grain element will be calculated utilizing the Gauss Legendre quadrature over the plane strain element regions [20,21], while the body force from grain boundary elements can use the analytical integration directly. They are:

For the grain part $P_{CG}$ [20,21,23]:

$$P_{CG} = \iint [B_G]^T \cdot [D_G] \cdot \varepsilon_{CG} \, dxdy, \tag{1}$$

$$P_{CG} = \sum_{i=1}^{nip} [B_G]^T \cdot [D_G] \cdot \varepsilon_{CG} \cdot W_i, \tag{2}$$

where the $[B_G]$ is the displacement–strain matrix of grain element, $[D_G]$ is the stress–strain matrix of grain element, and $\varepsilon_{CG}$ is the creep strain, nip is the number of the Gauss point and $W_i$ is the weighting coefficient.

For the GB part $P_{CGB}$:

$$P_{CGB} = \int [[B_{GB}] \cdot [T]]^T \cdot [D_{GB}] \cdot U_{GB} \, dl, \tag{3}$$

$$P_{CGB} = [[B_{GB}] \cdot [T]]^T \cdot [D_{GB}] \cdot U_{GB} \cdot L, \tag{4}$$

where the $[B_{GB}]$ is the node-element displacement matrix of grain boundary element, $[T]$ is the local–global coordination transfer matrix of grain boundary element, $[D_{GB}]$ is the stress—relative displacement matrix of grain boundary element, $U_{GB}$ is the creep jump displacement of the grain boundary, and $L$ is the length of the joint element.

Therefore, the global body loads can be obtained

$$P_C = P_{CG} + P_{CGB}, \tag{5}$$

where $P_C$ is the global body loads, $P_{CG}$ is the global body loads of grain part, $P_{CGB}$ is the global body loads of grain boundary part.

### 2.2. Numerical Implementation of GB

As mentioned before, the mesh of the GB part is implemented by the Goodman element [18]. Therefore, the core of the simulation of the GB is to obtain the element stiffness matrix of the Goodman element.

### 2.2.1. Goodman Element Stiffness

The Goodman element is a four-node with an eight-DOF element without thickness (as shown in Figure 2), initially, the nodal pairs (1,2) and (3,4) lie together under the unloading condition. The deformation of the element is represented by the relative displacement of the upper and lower surface.

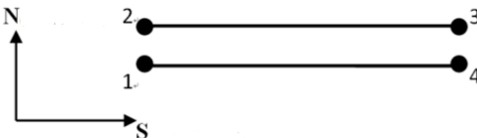

**Figure 2.** Goodman element schematic figure. N for normal direction. S for separate direction.

The derivation process of the element stiffness matrix is as follows [18,24]:

The displacement on separate direction for each node is $v_1$, $v_2$, $v_3$, $v_4$, while the displacement on the normal direction is $u_1$, $u_2$, $u_3$, $u_4$.

The shape functions for this four-node Goodman element with corner nodes take the form of (Equation (6))

$$\begin{aligned} N_1 = N_2 = 1 - \tfrac{2x}{L}, \\ N_3 = N_4 = 1 + \tfrac{2x}{L}, \end{aligned} \tag{6}$$

The displacement for the upper and lower surface of the element

$$[\Phi] = \begin{bmatrix} u^u - u^l \\ v^u - v^l \end{bmatrix} = \frac{1}{2} \cdot [B] \cdot \begin{bmatrix} u_1 \\ v_1 \\ u_2 \\ v_2 \\ u_3 \\ v_3 \\ u_4 \\ v_4 \end{bmatrix}, \tag{7}$$

$$[B] = \begin{bmatrix} -N_1 & 0 & N_1 & 0 & N_3 & 0 & -N_3 & 0 \\ 0 & -N_1 & 0 & N_1 & 0 & N_3 & 0 & -N_3 \end{bmatrix}, \tag{8}$$

The vector of the forces on unit length of the Goodman element at normal and separate directions are

$$[F] = \begin{bmatrix} F_n \\ F_s \end{bmatrix}, \tag{9}$$

$$\begin{bmatrix} F_n \\ F_s \end{bmatrix} = [D] \cdot [\Phi], \tag{10}$$

Here [D] denotes the stiffness matrix and have the form $\begin{bmatrix} k_n & 0 \\ 0 & k_s \end{bmatrix}$. [$\Phi$] denotes the element displacement (the details are shown in Equation (7)).

Based on the potential-energy theory, the element deformation energy P can be obtained as

$$P = \frac{1}{2} L \Phi^T [K] \Phi = \frac{1}{2} \int_{-\frac{L}{2}}^{\frac{L}{2}} \frac{1}{4} [\Phi]^T [B]^T [D] [B] [\Phi] dx, \tag{11}$$

Here, [K] is the stiffness matrix of the Goodman element per unit length.
Based on the calculation

$$[K] = \int_{-\frac{L}{2}}^{\frac{L}{2}} \frac{1}{4} [B]^T [D] [B] \cdot dx, \tag{12}$$

According to the Equations (10) and (11),

$$[B]^T[D][B] = \begin{bmatrix} N_1^2 K_S & 0 & -N_1^2 K_S & 0 & -N_1 N_3 K_S & 0 & N_1 N_3 K_S & 0 \\ 0 & N_1^2 K_N & 0 & -N_1^2 K_N & 0 & -N_1 N_3 K_N & 0 & N_1 N_3 K_S \\ -N_1^2 K_S & 0 & N_1^2 K_S & 0 & N_1 N_3 K_S & 0 & -N_1 N_3 K_S & 0 \\ 0 & -N_1^2 K_N & 0 & N_1^2 K_N & 0 & N_1 N_3 K_N & 0 & -N_1 N_3 K_S \\ -N_1 N_3 K_S & 0 & N_1 N_3 K_S & 0 & N_3^2 K_S & 0 & -N_3^2 K_S & 0 \\ 0 & -N_1 N_3 K_N & 0 & N_1 N_3 K_N & 0 & N_3^2 K_N & 0 & -N_3^2 K_S \\ N_1 N_3 K_S & 0 & -N_1 N_3 K_S & 0 & -N_3^2 K_S & 0 & N_3^2 K_S & 0 \\ 0 & N_1 N_3 K_N & 0 & -N_1 N_3 K_N & 0 & -N_3^2 K_N & 0 & -N_3^2 K_S \end{bmatrix} \tag{13}$$

The integrals of these $N_1^2$, $N_3^2$ and $N_1 N_3$ are

$$\int_{-\frac{L}{2}}^{\frac{L}{2}} N_1^2 = \frac{4}{3} L,$$
$$\int_{-\frac{L}{2}}^{\frac{L}{2}} N_3^2 = \frac{4}{3} L, \tag{14}$$
$$\int_{-\frac{L}{2}}^{\frac{L}{2}} N_1 N_3 = \frac{2}{3} L,$$

Thence, the [k] is

$$
K = \frac{1}{6} * \begin{bmatrix}
2K_S & 0 & -2K_S & 0 & -K_S & 0 & K_S & 0 \\
0 & 2K_N & 0 & -2K_N & 0 & -K_N & 0 & K_N \\
-2K_S & 0 & 2K_S & 0 & K_S & 0 & -K_S & 0 \\
0 & -2K_N & 0 & 2K_N & 0 & K_N & 0 & -K_N \\
-K_S & 0 & K_S & 0 & 2K_S & 0 & -2K_S & 0 \\
0 & -K_N & 0 & K_N & 0 & 2K_N & 0 & -2K_N \\
K_S & 0 & -K_S & 0 & -2K_S & 0 & 2K_S & 0 \\
0 & K_N & 0 & -K_N & 0 & -2K_N & 0 & 2K_N
\end{bmatrix}
\tag{15}
$$

### 2.2.2. Coordinate System Transmission

The previous consideration is a special case, which is the local coordinate that coincides with the global coordinate system. Therefore, it is necessary through the transmission matrix to relate these two coordinate systems (the geometry is shown in Figure 3).

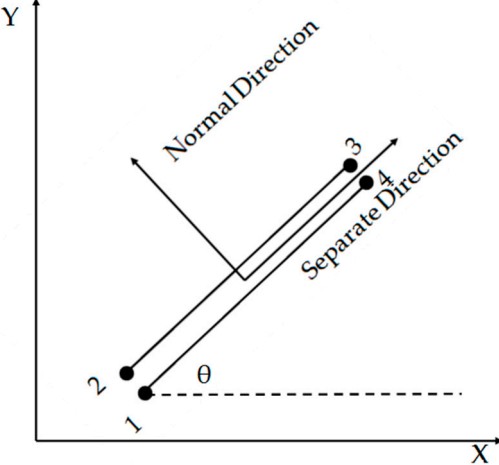

**Figure 3.** The schematic of the Goodman element in global coordinate system.

The rotation matrix is introduced in this procedure [24].

$$
T = \begin{bmatrix}
N & 0 & 0 & 0 \\
0 & N & 0 & 0 \\
0 & 0 & N & 0 \\
0 & 0 & 0 & N
\end{bmatrix},
\tag{16}
$$

where

$$
N = \begin{bmatrix}
\cos\theta & \sin\theta \\
-\sin\theta & \cos\theta
\end{bmatrix},
\tag{17}
$$

Based on the matrix [T], the stiffness matrix of the Goodman element can be obtained by

$$
[K]_{Glo} = [T]^T [K]_{loc} [T],
\tag{18}
$$

### 2.3. Programming Implementation of GB

The structured programming method is used to improve programming efficiency, and a mature overall structure and related standard subroutines were adopted, as mentioned at Section 2.2. In addition, some subroutines and structures have been developed to achieve material micro-creep simulation. These developments are based on the mathematical theory which were described in

Section 2.3, the function is to acquire the GB element stiffness matrix. The introduction and description of these subroutines are summarized in Table 2.

**Table 2.** Developed subroutines for GB simulation.

| Name | Arguments | Description |
|---|---|---|
| element_inf | length, angle, coord2 | Returns the element information: length ('length') and the angle θ ('angle'). The import information is the coordination of these four nodes under the global system, which store in array 'coord2'. |
| Loc_Gol | angle_t, angle | Returns the rotation matrix 'angle_t', for the transmission of the Goodman element form local to global coordination system. The import information is the angle θ which is generated by subroutine 'element_inf'. |
| new_km | km, kcoh | Returns the stiffness matrix 'km', 'km' is $8 \times 8$ size matrix, for Goodman element with the normal (n) and separate (s) rigidity of the Goodman element. The information is shore in 'kcoh' array. The mathematical background is Equation (22). |

The specific process for obtaining the Goodman element stiffness matrix, is: firstly, initialize the variables of each Goodman element, such as the rotation angle, the length and the element stiffness matrix, then based on the mathematical background which had been mentioned in Section 2.2 to obtain the stiffness matrix of single element. Finally, the 'element-by-element' and 'lower triangular' techniques were used to assemble the global stiffness matrix and to store the global matrix as a skyline vector respectively.

### 2.4. Creep Constitutive Equation

#### 2.4.1. Grain Element

Power law creep was chosen for the grain element in the current version of the software. Its uni-axial version is

$$\dot{\varepsilon} = A \cdot \exp\left(-\frac{Q}{RT}\right)\sigma^n t^m, \tag{19}$$

The multi-axial form is

$$\dot{\varepsilon} = A \cdot \exp\left(-\frac{Q}{RT}\right)\sigma^n t^m \frac{3}{2}\frac{S_{ij}}{\sigma_e}, \tag{20}$$

where n and m are the stress and time-hardening exponents respectively, the Q is the activation energy, the R is the universal gas constant and the A is the constitutive constant.

#### 2.4.2. Grain Boundary Element

The GB displacement jump at a normal direction can be obtained by the model which is developed by Vöse et al. [25]. It takes into account nucleation, growth, coalescence, and sintering of multiple cavities, i.e.,

$$\frac{d\beta}{d\bar{t}} = \frac{3}{2}\frac{\beta}{\bar{\rho}}(\bar{\alpha}_p - \bar{\alpha}_a) + \sqrt{\bar{\rho}}\sqrt[3]{36h(\psi)\pi\beta^2}\frac{d\bar{a}}{d\bar{t}}, \tag{21}$$

$$\frac{d\bar{\rho}}{d\bar{t}} = \bar{\alpha}_p(1-f) - \bar{\alpha}_a, \tag{22}$$

$$\bar{\alpha}_a = x_3 \cdot 8\pi\bar{\rho}^2\bar{a}\frac{d\bar{a}}{d\bar{t}}, \tag{23}$$

$$\frac{d\bar{a}}{d\bar{t}} = x_1 \cdot \frac{2\bar{D}_{gb}}{h(\psi)}\frac{\left[1 - \bar{a}_{tip}(\bar{a})\cdot(1 - x_2\omega)\right]}{\bar{a}^2 \cdot q(x_2\omega)}, \tag{24}$$

$$\omega = \sqrt[3]{\frac{9\pi\beta^2}{16h^2(\psi)}}; \bar{a} = \frac{1}{\sqrt{\bar{\rho}}}\sqrt[3]{\frac{3}{4}\frac{\beta}{h(\psi)\pi}}, \tag{25}$$

$$f = \frac{(\eta - 1)\omega}{1 - \omega}, \tag{26}$$

$$\eta = \exp\left(\left[x_4 \cdot 2\pi \overline{D}_{gb}\left(\overline{a}_{tip}(\overline{a} = 1) - \overline{a}_{tip}(\overline{a})\right)\overline{\rho}\left(\frac{d\overline{\mu}^p}{d\overline{t}}\right)^{-1}\right]\right), \tag{27}$$

$$\frac{d\overline{\mu}^p}{d\overline{t}} = \frac{\beta}{\sqrt{\overline{\rho}^3}}(\overline{\alpha}_p - \overline{\alpha}_a) + \sqrt[3]{36h(\psi)\pi\beta^2}\frac{d\overline{a}}{d\overline{t}}, \tag{28}$$

where

$$q(\omega) = -2\ln\omega - (3 - \omega)(1 - \omega); \overline{a}_{tip}(\overline{a}) = 2\overline{\gamma}_s \sin\psi/\overline{a}, \tag{29}$$

In this equation, $\beta$ is the damage variable, $\rho$ is the cavity density, a is the average radius of the cavity, $\overline{\alpha}_p$ is the stress dependent nucleation rate, $\overline{\alpha}_a$ is the annihilation rate, $\psi$ is the dihedral angle of the cavity (70°), $\overline{D}_{gb}$ is the GB diffusion coefficient, and $\omega$ is the damaged area fraction. The creep degradation of GB is calibrated by three variables: $\rho$, $\beta$, and a. These three parameters not only determine the failure degree of GB, but also determine the amount of the creep non-linear deformation. Therefore, $\rho$, $\beta$, and a are the three indicators for the benchmark.

Unlike grains, the deformation amount of GB is quantified by the absolute relative displacement of the upper and lower surface, it is assumed that this displacement is related to the stress and damage, but not rate related. The absolute relative displacement of the GB is determined by two variables, $\rho$ and $\beta$.

$$D_C = \frac{\beta}{\sqrt{\rho}} - \frac{\beta_0}{\sqrt{\rho_0}}, \tag{30}$$

where the $D_C$ is the relative displacement of creep, $\beta_0$ is the initial damage value (here is $10^{-4}$ and $\rho_0$ is the initial cavity density (here is $10^{-3}$ mm$^{-2}$). For the sliding part of grain boundary, the Newtonian viscous flow [15] is used

$$\frac{du_{sliding}}{dt} = \frac{\sigma_{sliding}}{\eta_{slding}}, \tag{31}$$

where $\frac{du_{sliding}}{dt}$ is the relative sliding velocity, $\sigma_{sliding}$ is the stress at the separate direction, $\eta_{slding}$ is the sliding viscosity of the GB.

### 2.4.3. Parameters for Constitutive Equation

This bi-grain case study simulates the micro creep process of copper at 500 °C. As mentioned before, power law creep is used to describe the creep mechanism of the grain part.

The material parameters of copper power law for grain [26] are (400–700 °C)

$$A: 38.8 \text{ MPa}^{-n}S^{-m-1}, Q: 197 \text{ KJ·mol}^{-1}, n: 4.8, m: 0.$$

The three main normalized material parameters for the grain boundary creep evolution (Markus Vöse's equations) of Copper at 500 °C are listed as [25]

$$\sqrt[3]{\overline{D}_{gb}}: 3.9695, \overline{\gamma}_s: 0.089, \overline{\alpha}_p: 0.24, \overline{R}: 42.$$

### 3. Preliminary Validation of the In-House Procedure

In order to make the logic and efficiency of the verification, a general step is employed [20,23,27]. The main idea of the step is from the uni-axial condition to the multi-axial condition, from linear elasticity to creep nonlinearity. The bi-grain case is chosen as the first step of the benchmark [17], which is to demonstrate the application of the in-house procedure under the uni-axial loading condition. In the first case, no shear sliding part occurs in the simulation due to the stress on the separate direction

being zero. Based on this case, the numerical stability and accuracy of the procedure is verified to pave the way for the subsequent polycrystalline simulation. In the second case, shear sliding occurs in the simulation GB to validate the accuracy and stability of the procedure on the separate direction.

### 3.1. Validation of Stress Update

The bi-grain consists of two square grains and a single GB as depicted in Figure 4.

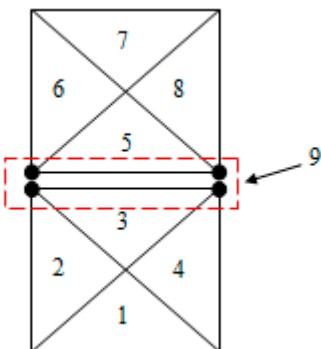

**Figure 4.** Schematics showing the bi-grain case study.

The model is meshed by four triangle plane strain elements for each the grain (e.g., 1–4, and 5–8, respectively), and 1 Goodman element for the GB. A uniformly distributed loading 20 MPa is applied on the top surface in the $Y$ direction and left line fixed on the $X$ direction and bottom line fixed on the $Y$ direction. In this case, no shear sliding stress occurs along the GB.

### 3.1.1. Validation of the Elastic Stress

The theoretical stress of the grain elements at the Gauss point is 20 MPa at $Y$ direction and the stress in the $X$ direction and the shear stress should be zero. The theoretical stress of the GB is 20 MPa at $Y$ direction and the stress in the $X$ direction should be zero. By analyzing the elastic simulation results of these nine elements (eight elements for the grain part and one element for the GB part): (1) For the grain part, the simulation results is shown in good agreement with the theoretical result at $Y$ direction. The maximum stress in $X$ direction is $1.066 \times 10^{-14}$ MPa (No. 2 grain element) and in shear direction is $4.170 \times 10^{-15}$ MPa (No. 6 grain element), which is negligible as expected. (2) For the GB part, the elastic stress at normal direction is 20.000 MPa and separated direction is $-2.033 \times 10^{-15}$ respectively, therefore, the simulation result is expected.

### 3.1.2. Validation of the Non-linear Creep Iteration

The solution to the creep boundary problem is based on the non-linear iterations. Therefore, the accuracy of the elastic stress field which was generated by the iteration directly affects the integral of the creep constitutive equations. Under the uniform stress field, since the creep deformation of each element is the same, the creep body loads are equal, and there is no stress redistribution. In summary, the convergence sign of the iteration is that the elastic stress at each iteration step is to keep constant, the bi-grain case belongs to this category. The iterative process lasts for 141,564 steps, with the time step of 0.0000001 (normalized). Through the observation of the elastic stress at each iteration step, it is clear the applied stress for the non-linear iteration of each element at every step keep constant. At the end of the iteration, the cumulative maximum error value of the stress for grain part are $-1.790 \times 10^{-11}$ MPa (No. 7 grain element) and $-1.69$ MPa (No. 8 grain element) for $X$ direction and shear direction respectively. For the GB part, the applied stress at the end of the simulation is 0 MPa at separated direction and 20.000 MPa at normal direction. Therefore, the in-house procedure has shown good convergence in the creep non-linear iteration.

### 3.1.3. Validation of the Integration

The accuracy verification of the non-linear iteration has been reported previously. Therefore, in this section, it reports the validation of the integration part. In the current version, the Euler forward method is adopted here for the integration of the constitutive equation (it was mentioned in Section 2.4.2).

### 3.1.4. The Result of the Bi-Grain Case Study

As mentioned before, there are three important indicators of the constitutive model: the cavity density $\rho$, the dimensionless damage variable $\beta$ and the average radius cavity a. The result of these three indicators versus the normalized time is plotted in Figure 5.

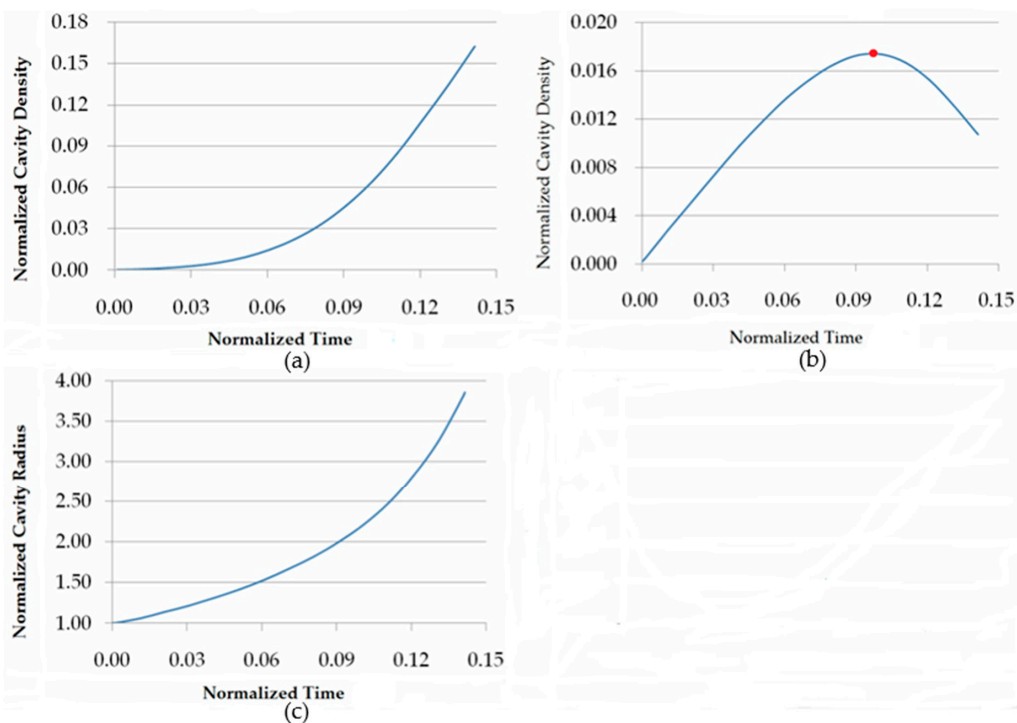

**Figure 5.** Cavitation evolution of these two important indicators. (**a**) The change of the damage variable $\beta$ versus the normalized time. (**b**) The change of the cavity density (normalized) versus the normalized time. (**c**) The change of the average radius a (normalized) versus the normalized time.

The rupture time is 0.142 (normalized), the maximum value of the $\beta$ is 0.162, the final value of $\rho$ is 0.011. In Figure 5a, it shows the evolution of the cavity density on the GB, it also reflects the variation of the cavity on the GB. The process can be divided into two stages: from the beginning, the quantity of the cavity grows, and when it reached the upper limit 0.017 at the time point 0.097 (normalized), cavity density decrease. The main reason for the reduction of the cavity density to the end is that random nucleation occurs on the GB throughout the process. However, cavity coalescence happens when the distance between two cavities is less than the critical value (0.1 times the initial cavity radio, here is 1 $\mu$m), when the quantity of the cavities reaches a maximum, the nucleation rate is less than the coalescence rate, eventually resulting in the decrease of the cavity density. Although the cavity density is reduced, the average radius of the cavity is increased continuously (shown in Figure 5c). Finally, the total volume of the cavities is increased, and the macroscopic phenomenon is that the creep jump displacement of the GB is increased.

### 3.1.5. Error Analysis

The mathematical simulation result of the constitutive equations (mentioned in Section 2.4.2) have been given in the article [25], although the results obtained does not involve finite element calculations, it can still be used in the benchmark of the bi-grain case study, since it is a special uniaxial loading condition. Since the exact value of the simulation is not given, it is unavoidable that there is a corresponding reading error. The percentage error between reading values and the simulation result is shown in Table 3. Based on the bi-grain case study, the result obtained applying the in-house procedure is shown in good agreement with the reference result. Based on this result, it demonstrates the numerical stability and accuracy of the procedure at the uniform stress condition. It also proves the accuracy of the subroutines for the numerical integration of the cavitation evolution model, and the convergence of the non-linear iteration is preliminarily verified under the non-sliding condition. The next bi-grain case will be set to verify the numerical accuracy of the sliding part.

**Table 3.** Percentage error of these five important indicators.

| Name | Reading Value [25] | Simulation Value | Percentage Error |
|---|---|---|---|
| Rupture Time (normalized) | 0.15 | 0.142 | 5% |
| Maximum value of the β | 0.165 | 0.162 | 1.53% |
| Final value of ρ (normalized) | 0.001 | 0.011 | 5.95% |
| The value of the ρ at the change point (normalized) | 0.017 | 0.017 | 2.11% |
| The time point of the change point (normalized) | 0.095 | 0.097 | 1.8% |

### 3.2. Validation of the Sliding Part

The FE model consists of two triangle grains and a single GB as depicted in Figure 6. The rotation angle of the joint element to the $X$ axial is 135°. The model is meshed by two triangle plane strain elements (for the grain part: each grain mesh by 1triangle element) and one Goodman element (for the GB part). In order to be logical and efficient, the normal stress of the GB is set to be consistent with the previous case (20 MPa). According to the geometric relationship, a uniformly distributed loading 40 MPa is applied on the top surface in the $Y$ direction and left line of the bottom grain element fixed on the $X$ direction and bottom line of the grain element fixed on the $Y$ direction. The main purpose of this case is to validate the accuracy of sliding deformation implementation and to benchmark the numerical accuracy of the Goodman element stiffness matrix in general condition (with angle). The geometric information of this FE model: the normalized length of the two short sides of the triangular grain is 1.

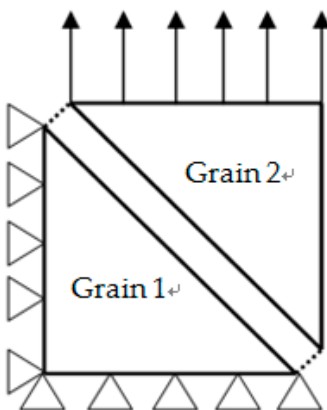

**Figure 6.** Schematics showing the second bi-grain case study to validate the sliding part.

### 3.2.1. Validation of the Elastic Stress

The theoretical stress of the grain elements at the Gauss point is 40 MPa at $Y$ direction and the stress in the $X$ direction and the shear stress should be zero. The theoretical stress of the GB is 20 MPa at normal direction and the stress in the separate direction should be −20 MPa. The elastic stresses for each element are shown in Table 4.

**Table 4.** Elastic stress for each element.

| Material Zone | Element No. | Direction | Elastic Stress (Unit: MPa) | Theoretical Value (Unit: MPa) |
|---|---|---|---|---|
| Grain | 1 | $X$ | $-3.553 \times 10^{-15}$ | 0 |
| | | $Y$ | 40.000 | 40 |
| | | $\tau$ | 0 | 0 |
| | 2 | $X$ | 0 | 0 |
| | | $Y$ | 40.000 | 40 |
| | | $\tau$ | $2.085 \times 10^{-15}$ | 0 |
| GB | 3 | Separate | −20.000 | −20 |
| | | Normal | 20.000 | 20 |

According to the Table 4, the simulation result from the procedure has been shown in good agreement with the theoretical value.

### 3.2.2. Validation of the Non-Linear Creep Iteration

As mentioned in the Section 3.1.2, according to Table 5, the in-house procedure has shown good convergence in the creep non-linear solution.

**Table 5.** The element elastic stress field at the selected iteration step.

| Iteration Step | Element No. | | Direction | Elastic Stress (Unit: MPa) | Theoretical Value (Unit: MPa) |
|---|---|---|---|---|---|
| 100,000 | Grain 1 | 1 | $X$ | $-9.623 \times 10^{-9}$ | 0 |
| | | | $Y$ | 40.000 | 40 |
| | | | $\tau$ | 0 | 0 |
| | Grain 2 | 2 | $X$ | $5.359 \times 10^{-10}$ | 0 |
| | | | $Y$ | 40.000 | 40 |
| | | | $\tau$ | $-9.812 \times 10^{-10}$ | 0 |
| | GB 1 | 3 | $X$ | −20.000 | −20 |
| | | | $Y$ | 20.000 | 20 |
| 231,704 | Grain 1 | 1 | $X$ | $4.134 \times 10^{-7}$ | 0 |
| | | | $Y$ | 40.000 | 40 |
| | | | $\tau$ | 0 | 0 |
| | Grain 2 | 3 | $X$ | $1.317 \times 10^{-8}$ | 0 |
| | | | $Y$ | 40.000 | 40 |
| | | | $\tau$ | $-1.349 \times 10^{-8}$ | 0 |
| | GB 1 | 5 | $X$ | −20.000 | −20 |
| | | | $Y$ | 20.000 | 20 |

### 3.2.3. Validation of the Sliding Part

The sliding model has been mentioned in Section 2.4.2. The magnitude $\eta_{sliding} = 3.85 \times 10^7 \text{Ns}/\text{mm}^3$ (normalized value is : 0.052) [10] of the sliding viscosity coefficient is chosen in this case. In this case, the elastic stress field keeps constant, and the sliding is a kind of linear model and the rupture time is 0.142 (normalized). Therefore, the absolute theoretical displacement of the sliding part is 2.721 (normalized). The simulation result value is 2.722. Therefore, the in-house procedure has shown good agreement with the theoretical value. Based on this result, it demonstrates the numerical stability and accuracy of the procedure at the bi-axial stress condition (normal and separate). It also proves the accuracy of the subroutines for the numerical integration of the sliding model. The convergence of the non-linear iteration with sliding deformation is preliminarily verified. The next step will be set to do a polycrystalline case study to benchmark that this procedure contains a potentially new application under the actual microstructure which considers the cavitation model and sliding model together.

## 4. Polycrystal Case Study

The plan strain simulation is conducted for copper–antimony alloy subjected to 10 MPa tensile stress at temperature is 823 K.

The geometry of this case is rectangle with 1 mm$^2$. The mesh for this model was generated by the Neper package [22]. Shape of the domain structure was built by the tessellation module (−T) of Neper and in a rectangular domain. The model contains 20 grains and 60 GBs. The mesh of the structure and re-mesh to generate the GB is done using the meshing module (−M) of Neper. The grains are meshed by 909 triangle plane strain elements and the GBs are meshed by 152 Goodman elements, as depicted in Figure 7 (GB is marked by the red line). The orientation of grain boundary elements is shown in Figure 8, which reveal the degree of reasonably random distribution.

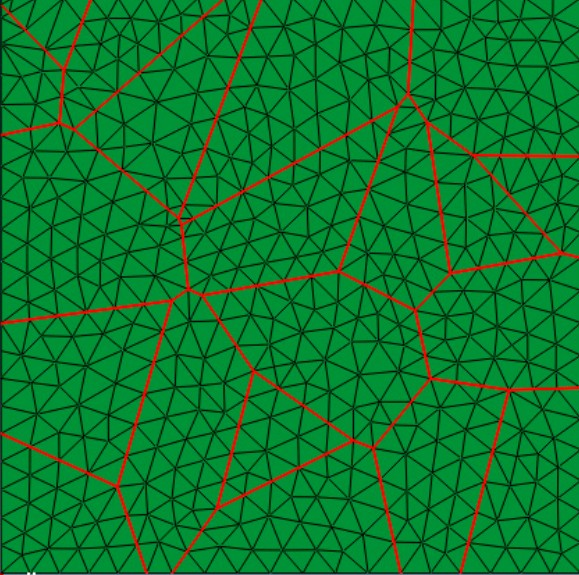

**Figure 7.** The mesh developed for the polycrystalline case study.

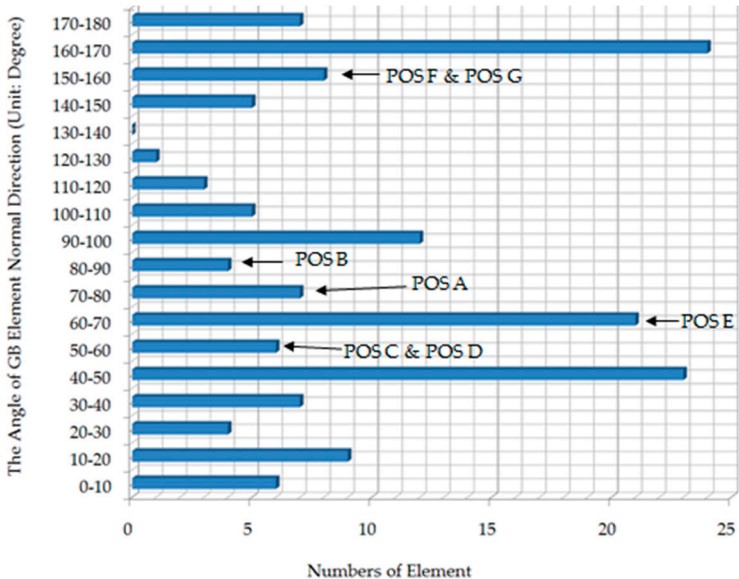

**Figure 8.** Orientation angle of the grain boundary element's normal direction.

In an attempt to conduct mesh convergence study, a series of much finer mesh has been produced, and it was found the computational time is too much and it is not realistic under current limitations of computational hardware to be vigorous. It is, unfortunately, to compromise to accept this size to proceed.

A uniformly distributed loading of 10 MPa was applied on the top surface in the $Y$ direction and left line and bottom line of the domain were fixed on the $X$ direction and the $Y$ direction, respectively.

In this case, it simulates the creep evolution of copper–antimony alloy at 823 K [11]. The parameters for the GB cavity model is: $\widetilde{D}_{gb} = 10^{-14} \text{mm}^5 \text{N}^{-1} \text{s}^{-1}$, $a_p = 2 \times 10^2 \text{mm}^{-2} \text{s}^{-1}$ and $b_p = 1$. The parameter for the copper power law creep has been mentioned in Section 2.4.3.

### 4.1. Rupture Time and Creep Damage Evolution

At the time of 78.9 h, there were seven grain boundary elements that failed. If that is deemed as creep rupture time, then it agrees with the majority of all uni-axial creep tests conducted [10]. In uniaxial test, the experimental condition is: copper–antimony alloy for material, loading is 10 MPa, and temperature is 823 K; one specimen fractured at 16.6 h, one broke at 17.9 h, one broke at 58.3 h. It is worth mentioning that the simulation was conducted for plane strain case, hence it is expected a longer lifetime at the same applied stress. The sequence of fracture and its time are shown in Figure 9 and Table 6.

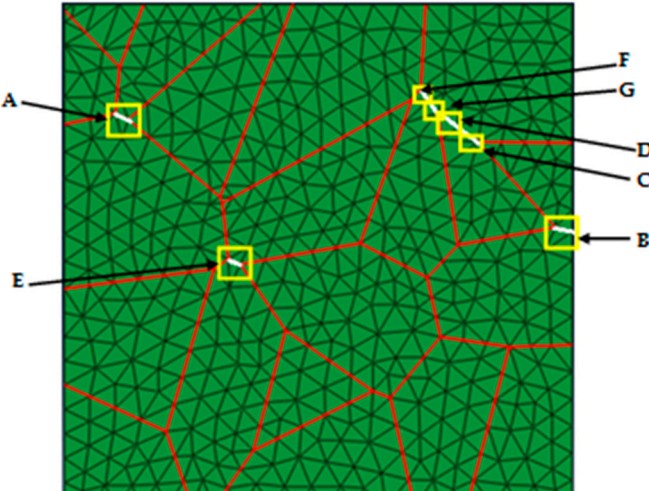

**Figure 9.** Sequence evolution of the failure position.

**Table 6.** Sequence and time of fracture of the first seven boundary elements.

| Position | Orientation Angle (Normal Direction) | Element No. | Time (Unit: h) | Step |
|----------|--------------------------------------|-------------|----------------|----------|
| A | 65.26084 | 48 | 23.55 | 12003387 |
| B | 76.16616 | 122 | 65.55 | 33246192 |
| C | 54.01357 | 93 | 68.48 | 34728834 |
| D | 54.01357 | 94 | 68.48 | 34728856 |
| E | 65.41204 | 111 | 70.69 | 35848560 |
| F | 146.3127 | 87 | 78.90 | 39987506 |
| G | 146.3128 | 88 | 78.90 | 39987517 |

The evolution of normal stress, cavitation rate, cavity density, and annihilation rate is shown in Figures 10–14, respectively.

From Figure 11, we can observe that the grain boundary A has the highest normal stress and stays relatively high until its fracture. The normal stress level in other failed elements is much lower than that is but still higher than the 10 MPa for most of the early part of the time. Hence, these elements could be subject to a higher nucleation rate and growth rate, finally resulting in failure. The initial higher normal stress clearly reveals the importance of grain boundary sliding and initial normal jumping in stress redistribution; and further creep deformation that redistributes the stress causing the decrease of high normal stress, but can still maintain itself for a longer period. Consequently, the damage developed in such a manner as confirmed by Figure 15.

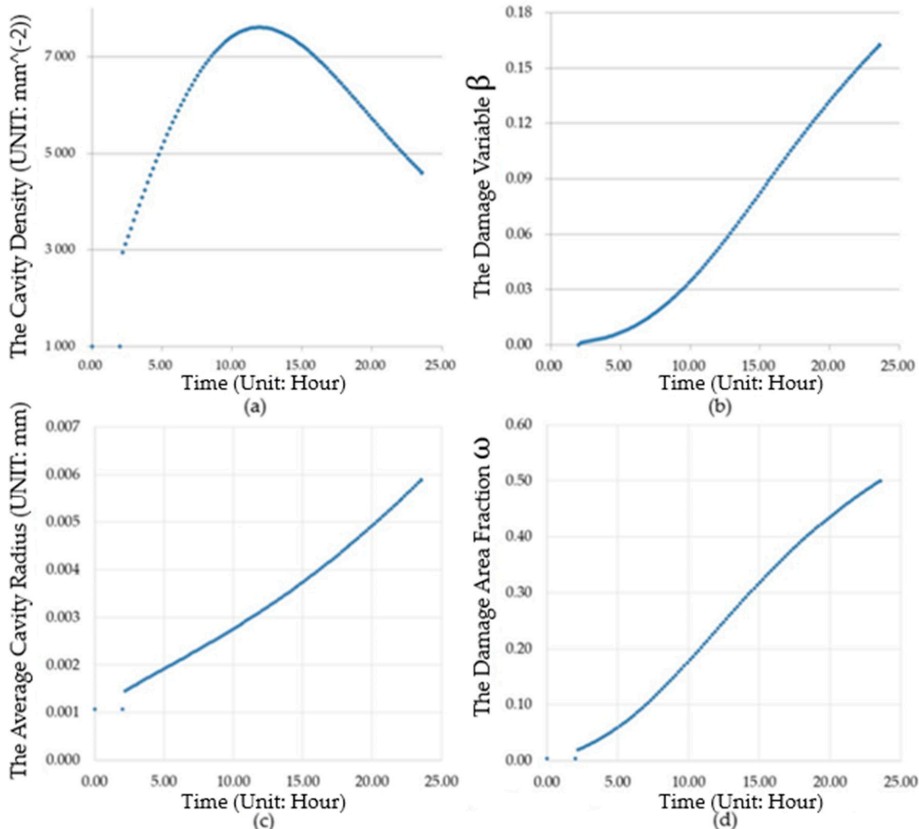

**Figure 10.** Evolution of the Position A. (**a**) The evolution of the cavity density versus the time. (**b**) The evolution of the average radius a versus the time. (**c**) The evolution of the average cavity radius a versus the time. (**d**) The damage area fraction versus the time.

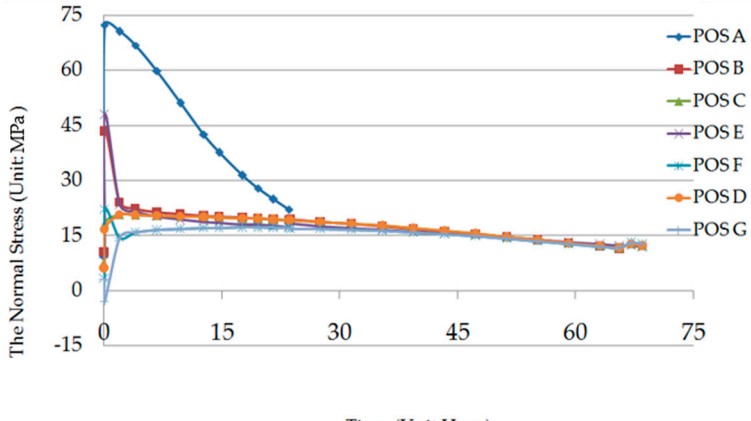

**Figure 11.** Evolution of the stress at normal direction.

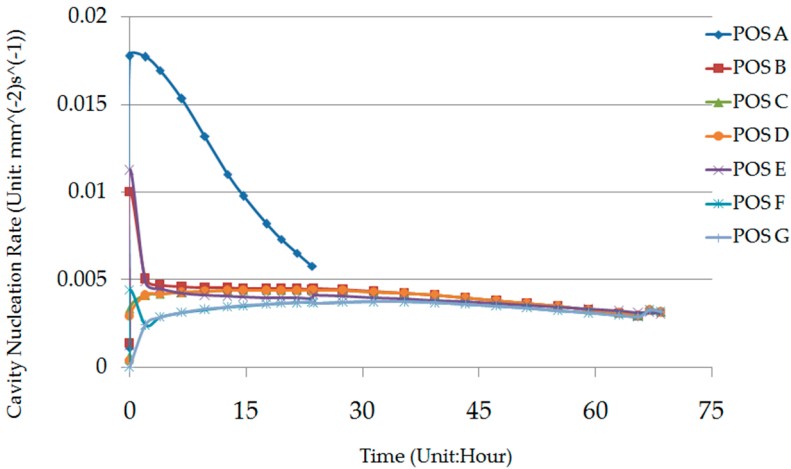

**Figure 12.** Evolution of the cavity nucleation ratio.

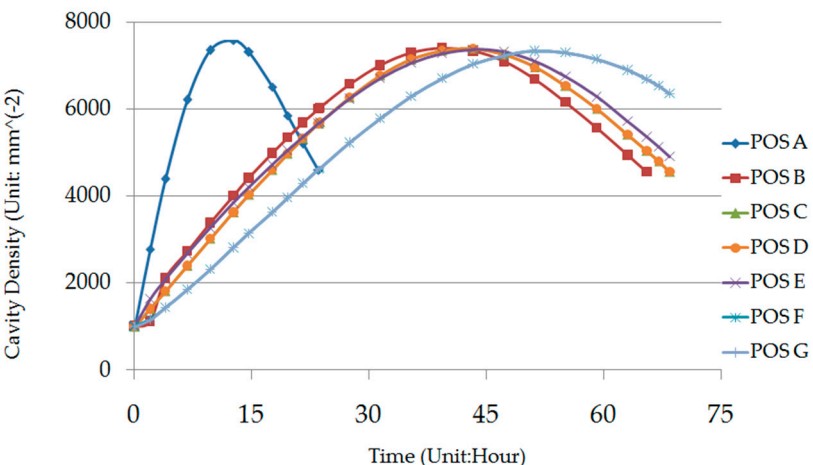

**Figure 13.** Evolution of the cavity density (POS B).

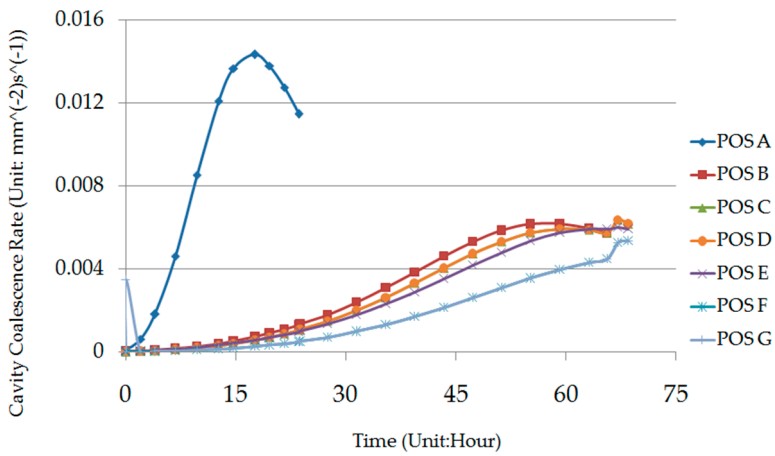

**Figure 14.** Evolution of the coalescence rate.

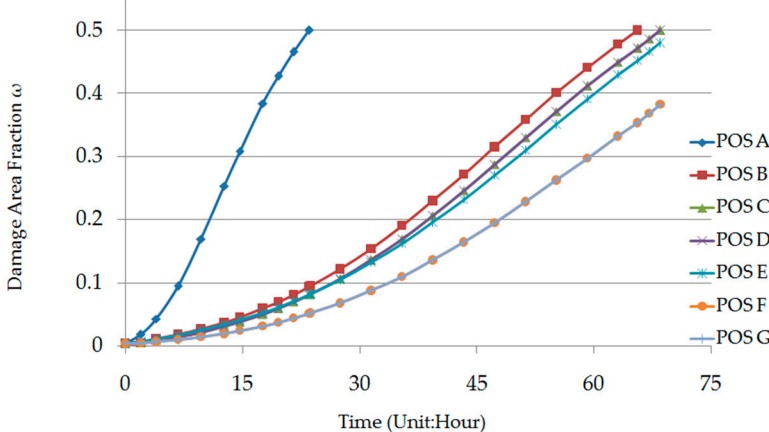

**Figure 15.** Damage evolution with time of the first seven failed grain boundary elements.

*4.2. Stress Field Evolution*

The normal stress evolution of all the grain boundary elements are shown in Figure 16a–f; and the creep damage are shown in Figure 17a–d, corresponding to the same time.

From the Figure 16c, it reveals that the damage is dominantly occurring at some slant degree to the direction of the axial stress, while the elements aligned with the direction of the stress were not damaged that much, see the range of elements in numbers 50–80. This may reveal that the normal node jump is relatively bigger than the sliding, so in this statistically under-terminated system, the normal stress has been released in those elements. This statistical trend clearly reveals the importance of grain boundary sliding and its effect on stress redistribution.

From the evolution of creep damage over time for all the elements, it also revealed another two facts: over the time, a reasonable portion of elements developed creep damage steadily and the component is about to rupture; hence it can be derived that the 152 grain boundary elements (its size and orientation distribution) did present the grain boundary fairly, though a study with finer mesh size will resolve this firmly.

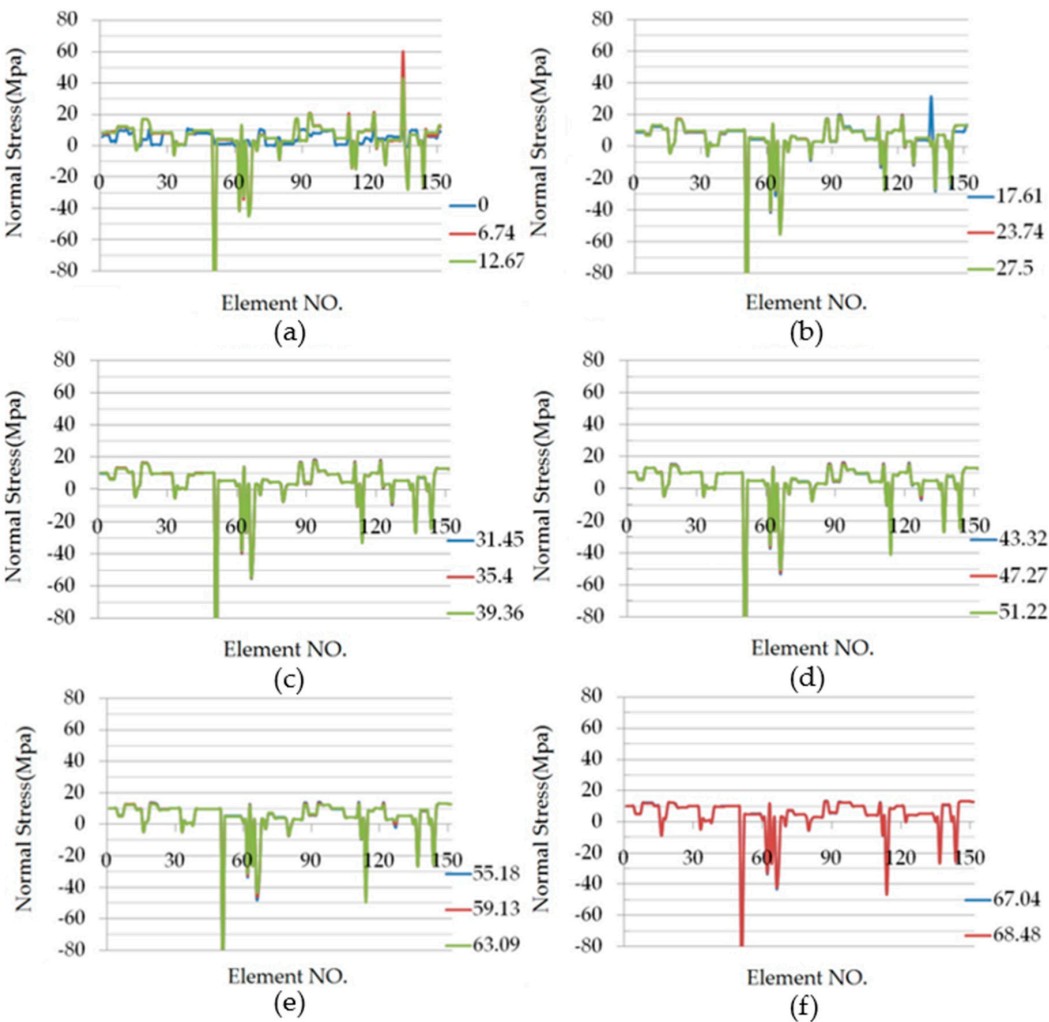

**Figure 16.** Evolution of normal stress of all grain boundary elements at the same time. (**a**) Time: 0 h, 6.74 h, 12.67 h. (**b**) Time: 17.61 h, 23.74 h, 27.5 h. (**c**) Time: 31.45 h, 35.4 h, 39.36 h. (**d**) Time: 43.32 h, 47.27 h, 51.22 h. (**e**) Time: 55.18 h, 59.13 h, 63.09 h. (**f**) Time: 67.04 h, 68.48 h.

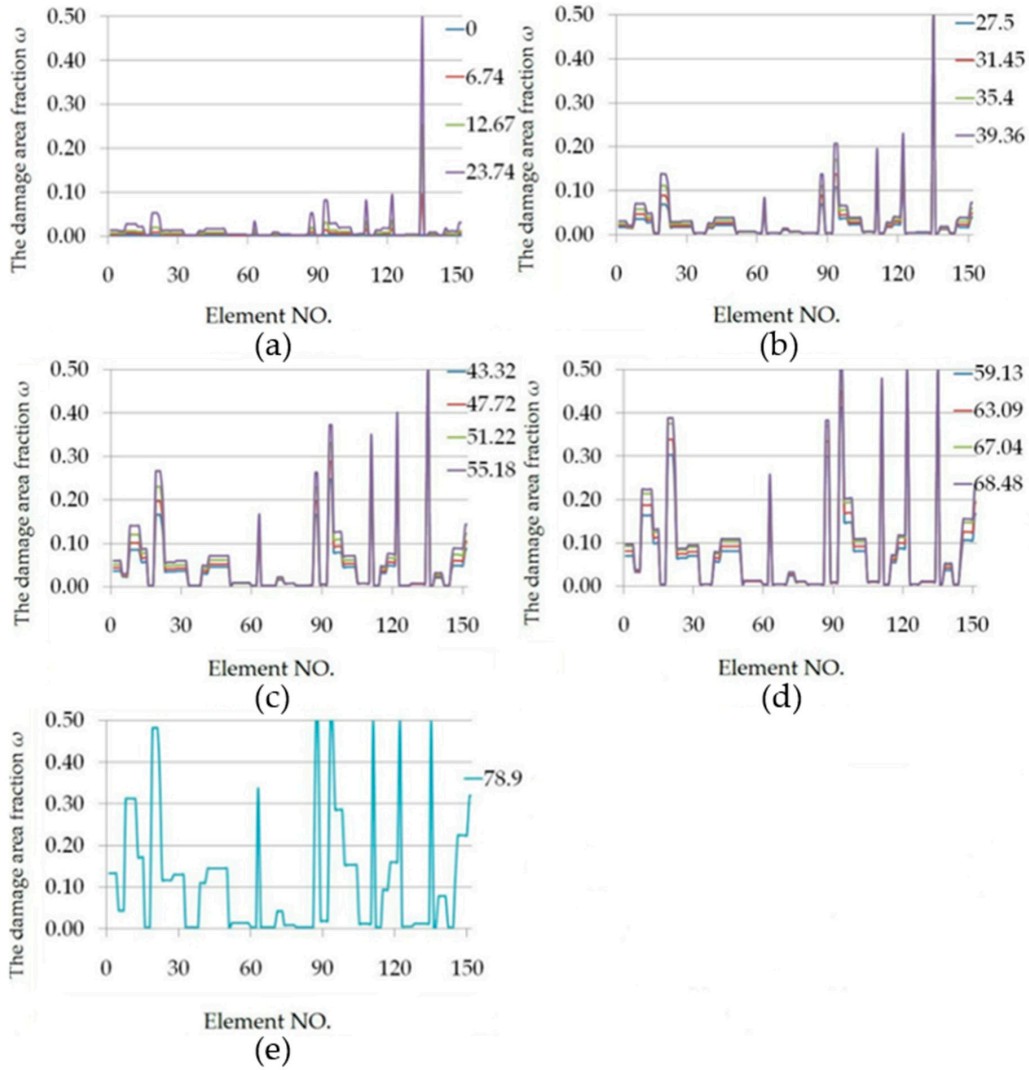

**Figure 17.** Damage evolution with time of the all the grain boundary elements at the same time.
(**a**) Time: 0 h, 6.74 h, 12.67 h, 23.74 h. (**b**) Time: 27.5 h, 31.45 h, 35.4 h, 39.36 h. (**c**) Time: 43.32 h, 47.27 h,
51.22 h, 55.18 h. (**d**) Time: 59.13 h, 63.09 h, 67.04 h, 68.48 h. (**e**) Time: 78.9 h.

## 5. Computational Cost

The three cases above was run in PC, its CUP is: Intel (R) Core (TM) i5-3337U 1.80 GHz.
The computational consumption (CPU time) of these three cases is shown in Table 7.

**Table 7.** CPU consumption time.

| Case | Consumption CPU Time (Unit: S) |
| --- | --- |
| Bi-grains (without sliding) | 6.36 |
| Bi-grains (with sliding) | 2.59 |
| Polycrystal (20 Grains) (first failure) | 598672 (6.9 days) |
| Polycrystal (20 Grains) (first seven failures) | 1995000 (23.1 days) |

Though the computer hardware is not high spec, it is still reasonable to claim that the computing
time and then the cost is not trivial if a full three-dimensional model is running. This justifies the
development and use of a two-dimensional model.

## 6. Conclusions and Future work

### 6.1. Conclusions

The development of and the preliminary application of two-dimensional finite element framework for creep damage simulation at grain boundary level was reported. It is concluded that:

1. The computational platform has been developed easily by adopting existing standard subroutines and/or algorithm. The computational time and cost are significant and higher spec computational hardware is desirable, if not necessary.
2. A simple plane strain simulating case revealed the stress concentration and its reduction.
3. The lifetime prediction still falls well within the experimental results. This is because the smeared-out grain boundary element is micro-mechanical based.
4. It also confirms that the modelling of grain is of second order of importance hence, some simplification is acceptable and justified.

### 6.2. Future Work

1. Conduct a parametric study to provide insight on the relative importance of various parameters throughout a lifetime;
2. Develop a three-dimensional version;
3. Develop and/or validate the micro-mechanical model using synchrotron micro-tomography cavity data.

**Author Contributions:** Q.X. conceived the idea to simulate the grain and grain boundary separately and preferred the in-house software development approach; Q.X. also participated in drafting, finalizing and submitting the paper; rewrote the abstract, introduction, discussion and future work for the revised version; advised what extra data/graphs (primarily Sections 4 and 5) to be produce in revised version and wrote the discussion; Q.X. finalized the paper and made submissions for all the three versions; J.T. proceeded with the theory, programming, and testing. J.T. also drafted the paper and participated in revising prior to its first submission, in addition to producing the data/diagrams, and provided some discussion regarding the results for the revised version; Z.L. participated in the initial drafting process.

**Funding:** This research received no external funding.

**Acknowledgments:** J.T. is grateful for being given this opportunity at the University of Huddersfield and for being awarded the Vice-chancellor PhD research scholarship. Q.X. is grateful for the Scholarship of Santander Student Mobility fund, The University of Huddersfield, 2017–2018.

**Conflicts of Interest:** The authors declare no conflict of interest.

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
