# Peer review of "Development of the FE In-House Procedure for Creep Damage Simulation at Grain Boundary Level"

_metals, doi:10.3390/met9060656_

Round 1
Reviewer 1 Report
Dear Authors,
You will find below some detailed suggestions. The paper has an innovative content that is not validated by mean of laboratory experimental activity.
line 9: “A 2-dimensional finite element framework….” instead of : “The 2-dimensional finite element framework…” .
line 10: This is a scientific paper not a project so, please check and make correction of the following sentence: “The rational for the project was that creep damage….”
Line 14: You state that: “ The conventional creep damage finite element framework for homogeneous material has been adopted and updated…” . It is not clear what you intend for “conventional creep damage finite framework”. Probably it is more simple and suggested to indicate that You have simulated the grain creep damage by mean of FE, or explain better.
Line 19: You state that: “ This paper contributes to the development of the finite element simulation for creep damage/rupture at a more realistic grain boundary level.” Probably you mean that “This paper is aimed to contribute for the development….”
Fig 4: it Is not necessary.
Tab.5: please use numbers with 2 o 3 significative decimals no more and declare that the numbers have been trunked
Line 384: contains a typing error.
More in general it is not clear how the model can be modified and adapted for , i.e.: steels, titanium, Ni-superalloys ets, etc. In fact I expect that the model should be revised and tuned for each one of the materials.
Author Response
Thank you and appreciate your valuable comments.
This revised paper has been proofread by a native English speaker.
All the suggested corrections have been accepted and implemented.
Reviewer 2 Report
The paper delas with the development of 2-dimensional finite element framework for creep damage simulation at grain boundary level.
The work is of interest for the journal but some critical point are present.
1) In the introduction must be clearified is the state of the art of creep damage simulation at grain
boundary level is at 2D simulation, or also 3D simulation have been proposed, but the 2D approach remains meaningful and valuable.
2) In the paper too many details of the standard FE formulation are reported, while detail and critical discussion of the implement model for the GB are missing ( is the sliding modelled?, is he thickness of the GB taken into accoount?) is the strain rate taken into account?
3) The model of creep inside the the grain is the base for the computation of the load at the GB, but a very simplified model has been used to model the creep of the grain. The effort to model the GB behaviour is justified, on this basis ?
If this assumption has been made just to perform the beanchmark of the GB subroutine it must be clerified in the paper, and also must be clarified if this assumption does not affect the beanchmark itself.
4) the validation of the proposed procedure is based on 2D B-grain while analogous reaserch based on implmentation of Gb creep damage in FE commercial Code presents simulation at the polycristal level, it must be justified with discussion about computation time (?) , simplicity in the model set-up (?), accuracy in the non linear solution?
5) Is the 2D B Grain with uniform normal stress and without sliding a valid benchmark case study? Please discuss.
6) The paper is hard to be read.
Remarks
- Line 99 "My specific work" is not appropriate
- Line 112-133 the reference [26] is reported too times. if the section is extract fro this references, it is enough to report the reference in the section title.
- Line125-127 are a repetition of Line 113-115
- Line 129 "Any reader who is interested to the original description or the complete list of the subroutines (80) is recommended to refer to the original document [26]." It is not appropriate for the paper and also it is obvious.
Line 140 "which was adopted by my colleague" It is not appropriate for the paper.
Line 154 "then to generate the creep body loads in the structure" It mens to generate the loads on the GB? Plase explain in he text.
Line 157-164 the inroduction to the GD subroutine is poor must be improved and made more clear.
Line 266 "The multiaxial form" please add "according to the Cooks and Ashby model"
Line 301 Plase specify the reference that has been assumed to verify the accuracy of the procedure.
Author Response
Thank you and appreciate your valuable comments.
This revised paper has been proofread by a native English speaker.
All the suggested corrections have been accepted and implemented.
Further questions raised by reviewer have been addressed, refer to the attached file.

Reviewer 3 Report
In my opinion authors must correct/clarify the following issues:
· The FE simulation must be described properly including details of the mesh, boundary conditions, type of contact, mesh convergence. Please explain how convergence can be achieved in a non-linear problem with a simulation considering just two elements. Besides define properly and explain deeply the contact conditions.
· In my opinion authors must include the results of the simulation shown in section 5 (future work) for making this paper interesting to readers.
#1) Line 10. Please use “paper” instead of “project”
#2) Line 10. This sentence is not clear, please revise it
#3) Line 14. This sentence is not clear, please revise it. Please be more precise.
#4) Line 15. Something seems to be missed in the sentence “and the division of grain boundary and grain boundary”?. I guess authors refers to “Grain and Grain Boundary”.
#6) Line 32. Something is missed in this sentence. Besides be more precise, which damage variable?.
#7) Line 33. Please delete “however”
#8) Line 37. Please use “(i)” instead of “1)” and “(ii)” instead of “2)” and so on
#9) Line 42. For the sake of clarity, give the selected one and add a reference
#10) Line 67. Please use “Object oriented finite (OOF)” instead of “OOF (Object oriented finite)”
#11) Line 85, 87 and 207. Please use “Goodman” instead of “goodman”
#12) Line 98. Please give a reference
#13) Line 99. Please give a reference
#14) Line 99. Please do not use first person in writing. The same in line 137.
#15) Line 112. Please do not use references in section titles.
#16) Line 126. Please give a reference
#17) Line 121-123. It is difficult to read this sentence, please rewrite it.
#18) Line 126. Please give a reference
#19) Line 131. Please provide a figure with higher quality to be readable.
#20) Line 148. Please use “applied” instead of “apply”
#21) Line 153. Please explain “mechanism of grain part”
#22) Line 153-156. Very confusing sentence, please revise it and rewrite it.
#23) Line 162. Please do not use “*” for scalar product, use “·” or a blank space
#24) Line 186. The quality of Fig. 2 is so poor please improve it. The same for Fig.3
#25) Line 270. Please for the sake of clarity, define all the parameters appearing in equations.
#26) Line 288. Something is missed in this sentence.
#27) Line 292. Sentence confusing. Please revise it.
#28) Line 308. Please explain deeply how the “theoretical stress of GB” is calculated
#29) Fig. 6 please place a label including the variable and unit represented in each axis of the plot. It is not clear how these results are calculated. Please explain them deeply.
#30) Line 341-342. Please justify how these results are obtained.
#31) Line 342. This sentence is confusing. Please revise it.
#32) Line 342. It is necessary to include the values with 15 decimals?
#33) Line 348. Please define “cavity radio”
#34) Line 359. It is not clear how these values are calculated. What is “readug values”
#35) Line 362. Please do not use “important”
#36) Line 358. What is “un-axial loading”?
Author Response
Thank you and appreciate your valuable comments.
This revised paper has been proofread by a native English speaker.
All the suggested corrections have been accepted and implemented.
Further question has been addressed in the attached file.

Round 2
Reviewer 1 Report
It is suggested to simplify without so many details (i.e: the programming lines in fig. 4. In fact the reviewer and the potential reader without the code and subroutines cannot check if it works or not. So it is suggested to explain how it work.
line 995: you state load, rupture time ...and what is the creep simulation temperature?
The creep is a complex phenomenon and the damage strongly depend upon several parameters and temperature is one of them, but I cannot see any reference to temperature that has been taken as base for your simulation.
Some minor suggestions: The Authors have just introduced the editing and minor improvements. Some editing mistakes still remain: line 1048 and line 1058.
In table IV, element 2, Y value is reported 20,000 . In the previous version the number was 1,9999999940162. Probably it is more correct to use only three decimals with the note that the values has been truncated.
The paper is still redundant with a lot of mathematical and programming details, sometimes not really useful. Moreover the original and valuable work seems to be at the beginning stage from a "metallurgical" point of view.
Author Response
Many thanks for your suggestions and we have updated the paper accordingly.

Reviewer 2 Report
I will thank the author for the improvement of the paper,
some more remarks are reported below:
1) "Recently, the X-ray micro-tomography has been used to investigate the cavitation of high Cr 171 steel [5-7], and the continuous cavity nucleation and cavity growth models were calibrated by Xu et 172 al [8] and an explicit creep fracture model, based on the coalescence of grain boundary cavity was derived [8]."
The X-ray micro-tomography procedureto to calibrate the model make the applicability of the model not general as declared at the end of introduction.
"the constitutive model is not mechanism based thus exists the issues 156 of low reliability and lack of general applicability."
Please revise the concept
2) "The applicability of Xu’s model to a wide range of stress [120 MPa to 180 MPa, lifetime 174 2825 to 51406 hours] has been demonstrated [9].
Actually 120-180 is not a wide range of stress.Please mitigate the sentence.
3) line 215-222 I suggest to remove these line because are not clear and not necessary.
4) I suggest to remove subsection by Introduction
5) The matematical fornulation is too long. some parts can be removed:
- line 309-319
-line 443-446
- line 400
- line 449-454
6) table 3 and Table 4 are too long please tray to compact the information
7) Line 1034 remove (Pos B)
8) Figure 16 and 17 are not necessary, please remove.
9) The presentation of future work can be include in the conclusion.
10) Any dependence on temperature is not discussed in the model. it is a paper on creep, some consideration must be done in general.
11) Line 114 A comparison can not be done if stres and temperature ofthe creep tests are not reported.
Author Response
Many thanks for your suggestions. We have updated the paper accordingly, except your suggestion to remove Figure 16 and Figure 17, which were requested by another reviewer, I am afraid.
Best regards
qiang

Author Response
Thank you for your review and support.
best regards
qiang
Round 3
Reviewer 1 Report
The paper has been properly revised